# “Single Knot–Single Running Suture” Vesicourethral Anastomosis with Posterior Musculofascial Reconstruction during Robot-Assisted Radical Prostatectomy: A Step-by-Step Guide of Surgical Technique

**DOI:** 10.3390/jpm13071072

**Published:** 2023-06-29

**Authors:** Rocco Simone Flammia, Eugenio Bologna, Umberto Anceschi, Antonio Tufano, Leslie Claire Licari, Luca Antonelli, Flavia Proietti, Federico Alviani, Michele Gallucci, Giuseppe Simone, Costantino Leonardo

**Affiliations:** 1Department of Maternal-Child and Urological Sciences, Sapienza University Rome, Policlinico Umberto I Hospital, 00161 Rome, Italy; roccosimone92@gmail.com (R.S.F.); antonio.tufano@uniroma1.it (A.T.); leslieclaire.licari@uniroma1.it (L.C.L.); luca.anto.92@gmail.com (L.A.); flavia.proietti@uniroma1.it (F.P.); federico.alviani@uniroma1.it (F.A.); costantino.leonardo@uniroma1.it (C.L.); 2Department of Urology, “Regina Elena” National Cancer Institute, 00128 Rome, Italy; umberto.anceschi@gmail.com (U.A.); puldet@gmail.com (G.S.)

**Keywords:** prostate cancer, robot-assisted radical prostatectomy, early continence, posterior musculofascial reconstruction

## Abstract

**Background:** Our aim is to describe Gallucci’s (VV-G) technique for vesicourethral anastomosis and posterior musculofascial reconstruction (PMFR) during robot-assisted radical prostatectomy (RARP) and to assess early urinary continence recovery and perioperative outcomes. VV-G consists of a “single knot–single running suture” vesicourethral anastomosis with PMFR. **Methods:** Between September 2019 and October 2021, we prospectively compared VV-G vs. conventional Van Velthoven anastomosis (VV-STD) during RARP. We excluded patients with urinary incontinence, pelvic radiotherapy, and urethral and BPH surgery prior to RARP. Social continence (SC) recovery, perioperative complications, and length of hospital stay (LOS) were compared between VV-G vs. VV-STD. SC was defined as 0–1 pad/die. We applied 1:1 propensity score matching (PSM) adjusting for different covariates (age, Charlson Comorbidity Index, BMI, prostate volume, nerve-sparing and lymph node dissection). **Results:** From 166 patients, 1:1 PSM resulted in two equally sized groups of 40 patients each with no residual differences (all *p* ≥ 0.2). VV-G yielded higher 3-month SC rates than VV-STD (97.5 vs. 55.0%, *p* < 0.001). A tiny difference was still recorded at one-year follow-up (97.5 vs. 80.0%, *p* = 0.029, HR: 2.90, 95% CI: 1.74–4.85, *p* < 0.001). Conversely, we observed no differences in any perioperative complications (15.0 vs. 22.5%, OR: 0.61, 95% CI 0.19–1.88, *p* = 0.4) and LOS (3 vs. 4 days, Δ: −0.69 ± 0.61, *p* = 0.1). **Conclusions**: VV-G significantly improved early SC recovery without increasing perioperative morbidity. In our opinion, VV-G represents an easy-to-learn and easy-to-teach technique due to its single-suture, single-knot, and symmetrical design.

## 1. Introduction

Prostate cancer (PCa) represents the second most common diagnosed cancer in men [1,2]. For decades, open radical prostatectomy has been the standard treatment of localized PCa [3]. Since its introduction in 2001, robot-assisted radical prostatectomy (RARP) has gained popularity [4]. Despite its increasing use, functional outcomes in terms of continence and erection still lag, markedly reducing the quality of life, especially for younger and more active patients [5,6]. Notably, the prevalence of urinary incontinence after RARP ranged from 4% to 31% [7]. In an attempt to improve continence rates, several technical refinements for RARP have been published [8]. Among them, posterior musculofascial reconstruction (PMFR) appears to play an essential role in early postoperative continence recovery [9,10]. Interestingly, a previous study on laparoscopic radical prostatectomy reported improved continence rates after ‘‘single knot–single running suture” vesicourethral anastomosis (VUA) with PMFR, using the Van Velthoven anastomosis technique modified by Gallucci (VV-G), vs. conventional Van Velthoven anastomosis without PMFR (VV-STD) [11]. However, these results have not been validated in the context of RARP yet. To address this void, we conducted an observational prospective study to describe VV-G during RARP. We hypothesized that this technique improves early social continence (SC) recovery without increasing perioperative morbidity in comparison with VV-STD.

## 2. Materials and Methods

### 2.1. Study Design

We performed an observational prospective study comparing VV-G vs. VV-STD during RARPs performed at our institution (Department of Urology, Policlinico Umberto I Hospital at Sapienza University in Rome, Rome, Italy) from September 2019 to October 2021. Data were prospectively collected within our institutional review board (IRB)-approved database for RARP-treated patients and written consent was obtained from all patients.

### 2.2. Study Population

We included patients aged < 75 years old with a histologically confirmed localized PCa (clinical stage T1c-2, N0, M0 according to AJCC 8th edition [12]) treated with RARP. We excluded all patients with urinary incontinence prior to surgery, history of pelvic radiotherapy and/or urethral surgery, urethral stricture, or BPH surgery prior to RARP.

### 2.3. Data Collection

Patient (age, height, weight, BMI, Charlson Comorbidity Index, smoking habits, diabetes) and tumor (pre-operative PSA, prostate volume, ISUP grade biopsy) characteristics were collected during hospitalization before surgery. Perioperative information was collected according to the operating sheet (nerve-sparing approach, pelvic lymph node dissection status) and medical records (postoperative complication according to Clavien–Dindo classification system [13], length of hospital stay). Pathologic data (pTNM, surgical margin status, pathologic ISUP grade, and surgical resection margins) were reported at final histopathologic examination. Functional outcomes were collected during outpatient visits within the first 12 months of follow-up. Continence status was investigated with personal interviews during outpatient visits and stratified according to number of pads/die: zero vs. one vs. multiple pads at 10 days, 1-, 3-, 6-, 9- and 12-months from catheter removal. To rely on a binary event for further analysis, the following SC definition was adopted: continent (0–1 pad/die) vs. incontinent (>1 pad/die) [14,15].

### 2.4. Endpoint

The primary endpoint was to describe the VV-G technique during RARP and to compare early SC recovery (3 months from catheter removal) between VV-G vs. VV-STD. The secondary endpoint was to compare perioperative outcomes (any complications and length of hospital stay) and one-year overall SC rates between VV-G vs. VV-STD.

### 2.5. Surgical Technique

Since September 2019, we began to progressively introduce VV-G as an alternative to VV-STD during transperitoneal RARP performed at our institution. The decision to perform VV-G vs. VV-STD depended on surgeon preference. The decision to perform extended pelvic lymph node dissection (ePLND) and/or nerve-sparing technique during RARP relied on patient and tumor characteristics in accordance with EAU recommendations [16,17]. All procedures were performed by a single expert surgeon (C.L.) at the plateau of his learning curve [18,19].

#### Anastomosis Phase Modified Sec. Gallucci

After the demolitive phase, the anastomosis running suture is prepared through binding together both ends of two 2-0 Monocryl sutures. VV-G anastomosis is performed considering the following steps: The first stitch is placed outside-in at the 6 o’clock position of the bladder neck—within the thickness of the detrusor muscle—taking care not to include the bladder mucosa (Figure 1A). The same step is performed with the opposite stitch; the “knot” formed by the two sutures joined together represents an anchoring point positioned on the posterior aspect of the bladder neck. The left suture is passed from right to left transversally through the posterior musculofascial aspect of the pelvic floor, incorporating remnants of the rectourethralis muscle, without including the urethral mucosa (Figure 1B), and then outside-in at the 7 o’clock position on the bladder neck following a conventional Van Velthoven anastomosis [20]. The left suture is then placed inside-out on the urethra at the 7 o’clock position and subsequently outside-in at the 9 o’clock position on the bladder neck. Similarly, the right suture is placed from left to right through the posterior musculofascial plate (Figure 1C), and then passed outside-in at the 5 o’clock position on the bladder neck, inside-out on the urethra at the 5 o’clock position, and subsequently outside-in at the 3 o’clock position on the bladder neck. The bladder neck is not moved towards the urethra until three passes on each side have been completed. When this is achieved, gentle traction is exerted on each suture simultaneously and alternately; the system of loops allows the approach between the bladder neck and the urethra without excessive traction, minimizing the possibility of injury to the anastomosis, configuring the posterior aspect of the anastomosis (Figure 1D). A silicone 20F catheter is then placed into the bladder. The left suture is now placed inside-out at the 10 o’clock position and, again, outside-in on the urethra at the 11 o’clock position. Subsequently, the suture is passed inside-out on the bladder neck at the 11 o’clock position. Similarly, the right suture is placed inside-out at the 2 o’clock position and, again, outside-in on the urethra at the 1 o’clock position (Figure 1E). Finally, after passing the right suture inside-out on the bladder neck at the 1 o’clock position, the ends of the two sutures are tied to one another on the outside part of the bladder (Figure 1F). The balloon on the 20F silicone catheter is filled with 10 mL of water. A drain is placed and is usually removed on the first postoperative day. The silicon catheter is routinely removed postoperatively on the 7th day without radiographic control.

### 2.6. Sample Size Calculations

Sample size calculations for the primary endpoint was computed. According to a previous study [21], a sample of at least 35 patients per group would allow to test 30% difference in urinary SC rates at 3 months after catheter removal, with α ≤ 0.05 and β = 0.10.

### 2.7. Statistical Analyses

Statistical analyses were based on the following steps. We stratified the overall cohort between VV-G vs. VV-STD. Descriptive statistics included frequencies and proportions for categorical variables and medians and interquartile ranges (IQR) for continuously coded variables. Pearson’s Chi-square test and the Wilcoxon rank sum test examined differences in proportion and median distribution, respectively. Due to the lack of randomization between VV-G and VV-STD patients, we performed 1:1 propensity score matching (PSM) adjusting for the following covariates: age, Charlson Comorbidity Index, BMI, prostate volume, nerve-sparing, and ePLND. All further analyses were performed in the PS-matched cohort. First, logistic regression and linear regression models addressed any postoperative complications and LOS, respectively. Second, the Fischer exact test examined 3- and 12-month SC rate differences. Since PMFR was expected to accelerate SC recovery, we applied generalized estimated equations (GEEs) for repeated measurements to estimate and to graphically depict the time-dependent probability of SC recovery within the first 3 months. Similarly, Kaplan-Meier plots and a Cox regression model addressed one-year overall SC recovery. All tests were two-sided with a level of significance set at *p* < 0.05, and the R software environment for statistical computing and graphics (version 3.4.3) was used for all analyses.

## 3. Results

### 3.1. Study Population Characteristics

Overall, we included 166 patients treated with RARP. VUA was performed according to the VV-G vs. VV-STD technique in 92 vs. 74 patients, respectively. VV-G patients harbored a slightly lower BMI (25.9 vs. 27.1, *p* = 0.026) than their VV-STD counterparts. Moreover, either a mono- or bilateral nerve-sparing approach (57.6 vs. 25.7%, *p* < 0.001) and ePLND (39.1 vs. 21.6%, *p* = 0.006) was performed more frequently in the VV-G vs. VV-STD group, respectively. Conversely, we observed no differences between VV-G vs. VV-STD, based on other patient and/or tumor characteristics (Table 1).

### 3.2. Propensity Score-Matching

To address population differences, PSM was applied between 92 VV-G and 74 VV-STD patients. One-to-one PSM (age, Charlson Comorbidity Index, BMI, nerve-sparing, ePLND) resulted in two equally sized groups of 40 VV-G vs. 40 VV-STD patients, with no residual statistically significant differences (all *p* ≥ 0.2, Appendix A).

### 3.3. Perioperative Outcomes

After PSM, we observed no differences in complication rates between VV-G vs. VV-STD (15.0 vs. 22.5%, OR: 0.61, 95% CI 0.19–1.88, *p* = 0.4). Specifically, no patient exhibited high-grade complications (Clavien-Dindo ≥ 3), neither in the overall nor in the PS matched cohort. Moreover, no differences in median LOS were recorded between VV-G vs. VV-STD (3 vs. 4 days, Δ: −0.69 ± 0.61, *p* = 0.1).

### 3.4. Early Social Continence

After PSM, VV-G yielded higher 3-month SC rates than the VV-STD technique (97.5 vs. 55.0%, *p* < 0.001). At GEE analysis, VV-G exhibited a protective role for early SC recovery (OR 5.69, 95% CI 2.47–13.09, *p* < 0.001, Figure 2).

### 3.5. One-Year Overall Social Continence

After PSM, one-year overall SC rates were 97.5 and 80.0%, respectively, for VV-G and VV-STD (*p* = 0.029). Kaplan–Meier plots (Figure 3) depicted a median time to continence of 1 vs. 3 months for VV-G and VV-STD, respectively. These differences translated into an HR of 2.90 (95% CI 1.74–4.85, *p* < 0.001) in favor of VV-G.

## 4. Discussion

In 2001, Rocco et al. introduced a reconstruction technique for the posterior aspect of the rhabdosphincter [22] during open radical prostatectomy, based on an anatomical study of the rhabdosphincter [23]. This technique consisted of a two-layered reconstruction using interrupted sutures, binding together the free edge of Denovilliers’ fascia and the posterior bladder with the posterior aspect of the rhabdosphincter and the posterior median raphe. The purpose of this technique was to provide posterior support to the sphincteric mechanism and prevent caudal retraction of the urethra.

Subsequently, in 2006, a study conducted by Rocco and colleagues showed that posterior musculofascial reconstruction significantly reduced the time to complete continence after radical retropubic prostatectomy (RRP) [24]. In 2007, the same authors described the application of the posterior reconstruction technique to transperitoneal laparoscopic radical prostatectomy (LRP) [25].

Adaptation of this procedure into robotic surgery was first reported by Coughlin et al. in 2008 [26]. Since then, many authors have adopted the prostatic musculofascial plate technique during RARP with the aim of improving early continence after radical prostatectomy [21,27,28,29]. Among different techniques proposed, Gallucci et al. described a modified Van Velthoven anastomosis during laparoscopic radical prostatectomy with excellent results [11]. Notably, VV-G anastomosis has never been described during RARP. In consequence, we conducted a prospective study to illustrate VV-G anastomosis in a step-by-step fashion and to assess early urinary continence recovery and perioperative outcomes.

Firstly, 166 patients were included in this observational prospective study and stratified according to anastomosis technique between VV-G (*n* = 92) vs. VV-STD (*n* = 74) with comparable patient and tumor characteristics. Nonetheless, the nerve-sparing approach and ePLND were more frequently performed in VV-G than VV-STD patients. This difference may be related to preoperative and/or intraoperative surgeon decisions, which relied on information about patient and tumor characteristics, some of which may have not been captured based on the current study design. To maximally reduce imbalances between the two groups due to lack of randomization, we performed a one-to-one propensity score matching adjusting for the most important predictors of continence recovery such as comorbidity load, BMI, nerve-sparing approach, and ePLND. Moreover, after PSM, no differences in perioperative outcomes, such as any complications and LOS, were observed between VV-G and VV-STD. Since worse perioperative outcomes might have negatively affected early continence recovery, the lack of difference in terms of any complications and LOS further supports the unbiased comparison between VV-G and VV-STD patients in the current study cohort.

Second, we observed higher early SC recovery rate in patients treated with a VV-G anastomosis compared to their VV-STD counterparts. Interestingly, GEE analysis estimated a five-fold increase in SC recovery after VV-G vs. the VV-STD technique. These results agree with previous findings supporting the efficacy of PMFR prior to VUA to improve early continence recovery after RARP. Moreover, when relying on the entire pre-planned one-year follow-up and using a different statistical methodology, we observed that VV-G patients exhibited shorter median time to SC recovery than their VV-STD counterparts (1 vs. 3 months, HR 2.90, *p* < 0.001). These findings are extremely important since they support the efficacy of VV-G to accelerate SC recovery through relying on a longer follow-up and a different statistical approach.

Taken together, VV-G technique improved early SC recovery, with half of patients achieving continence within the first 30 days. Moreover, we observed a difference in continence at one-year follow-up (97.5 vs. 80.0%, *p* = 0.029). To the best of our knowledge, five previous studies [29,30,31,32,33] investigated the role of PMFR during RARP, adopting the same SC definition (Table 2). Overall, 3-month SC rates ranged from 63 to 91% in the experimental arm (with PMFR) vs. 21 to 91% in the control arm (without PMFR). Interestingly, the largest study investigating the role of PMFR (N = 396), published by Tewari et al. [30], reported higher 3-month SC rates after PMFR vs. the control group (91 vs. 50%, *p* < 0.001), which is in agreement with our results (98 vs. 55%, *p* < 0.001). Conversely, our results are in contrast with previously published RCTs [29,32,33] with comparable sample size. Indeed, these RCTs failed to find statistically significant differences in 3-month SC rates between the two treatment arms. However, despite the lack of randomization, we relied on prospective study design and PSM to maximally reduce covariate imbalances between the two groups.

Nonetheless, it’s important to emphasize that the primary objective of the current study was not to test the absolute effect of PMFR; rather, we aimed at illustrating the implementation of VV-G anastomosis with robotic surgery and reporting its perioperative and functional outcomes while providing a comparison with VV-STD. Even though someone might argue that the differences between VV-G vs. VV-STD reported by Simone et al. during LRP would disappear when moving to robotic surgery due to better overall performance of RARP vs. LRP for functional outcomes [34], we showed that this difference still exists when performing RARP. Finally, moving to technical considerations, we firmly believe that VV-G reflects an easy way to perform PMFR together with VUA.

We believe that the main strength of our anastomosis technique with posterior musculofascial reconstruction lies in its simplicity and, consequently, reproducibility. After the initial two passages of the right and left sutures at the six o’clock position on the bladder neck, the “knot”—formed through binding together the two terminal ends of two 2-0 Monocryl sutures—is placed on the posterior aspect of the bladder neck. This “anchor point” enables fixation during the initial stages of anastomosis, allowing for traction on it without altering the length of the two sutures, making the first phase of the anastomosis more seamless.

Focusing on posterior musculofascial reconstruction, some differences arise between our anastomosis technique and those previously described. In various posterior reconstruction techniques—from Rocco’s technique to various modifications and new techniques introduced over the years [29,30,31,32,33]—most techniques not only involve a “dedicated” suture in the reconstitution of the posterior plate but also several steps to increase its stability and holding strength. In contrast, our technique involves a posterior reconstruction directly “linked” with the anastomosis. Two transverse passages from right to left—with the left suture—and from left to right—with the right suture—offer a fast and straightforward reconstitution of the posterior plate.

We believe that this technique offers effective posterior reconstruction, ensuring potential benefits in terms of continence recovery while maintaining a smooth execution during this phase.

The first traction applied to the anastomosis occurs after the initial three steps on each side, completing approximately 50% of the anastomosis. This initial traction facilitates the approximation of the bladder towards the urethra, allowing for the recovery of the length of the two sutures, and preparing for subsequent steps to complete the anastomosis.

To summarize, we identified three main advantages: (1) only one running suture; (2) only one knot; (3) symmetrical design. Under such premises, the surgeon can proceed uninterruptedly to complete the PMFR together with VUA through avoiding interruptions due to the use of multiple sutures and multiple knots. Moreover, the VV-G is realized entirely symmetrically, making it easy to learn as well as easy to teach.

Our study is not devoid of limitations. First and foremost is the lack of randomization. Nonetheless, prospective data acquisition and PSM maximally reduced imbalances between the two groups due to potential selection bias. Second, we relied on a limited sample size. However, after PSM, we compared two equally sized group of 40 patients (similarly in published RCTs [29,33]) and achieved the pre-planned sample size required.

**Table 2 jpm-13-01072-t002:** Studies investigating the role of posterior musculofascial reconstruction during RARP adopting the same definition of social continence recovery (0–1 pad/die).

Article	TOS	LOE **	N° of Sutures	Continence Definition	PMFR		3-mo Continence Rate (*p*-Value) PMFR Yes vs. No	12-mo Continence Rate (*p*-Value) PMFR Yes vs. No
					Yes	No		
**Current study**	**Prospective**	**2b**	**1**	**0–1 PAD**	**92**	74	98 vs. 55% (<0.001)	98 vs. 80% (0.029)
Tewari et al. (2008) [30]	Retrospective	3b	3	0–1 PAD	182	214	91 vs. 50 (0.001)	-
You et al. (2012) [31]	Retrospective	3b	2	0–1 PAD	28	31	89 vs. 71% (0.1) *	95 vs. 92% (0.7) *
Jeong et al. (2015) [32]	RCT	1b	2	0–1 PAD	50	45	90 vs. 91% (0.9) *	-
Salazar et al. (2021) [29]	RCT	1b	2	0–1 PAD	80	72	84 vs. 78% (0.2)	95 vs. 94% (0.6)
Sutherland et al. (2011) [33]	RCT	1b	2	0–1 PAD	47	47	63 vs. 81% (0.07)	-

TOS = type of study; LOE = level of evidence; RCT = randomized controlled trial; PMFR = posterior musculofascial reconstruction. * = *p*-value was not provided in the original study and calculated based on available proportions using an online calculator for Pearson chi-square test. ** = according to level of evidence [35]

## 5. Conclusions

In the current study, we described Gallucci’s technique (VV-G) for vesicourethral anastomosis with posterior musculofascial reconstruction during robot-assisted radical prostatectomy. We observed a significant improvement in early social continence recovery in favor of Gallucci’s technique without increasing perioperative morbidity. In our opinion, VV-G represents an easy-to-learn and easy-to-teach technique for vesicourethral anastomosis due to its single-suture, single-knot and symmetrical design.

## Figures and Tables

**Figure 1 jpm-13-01072-f001:**
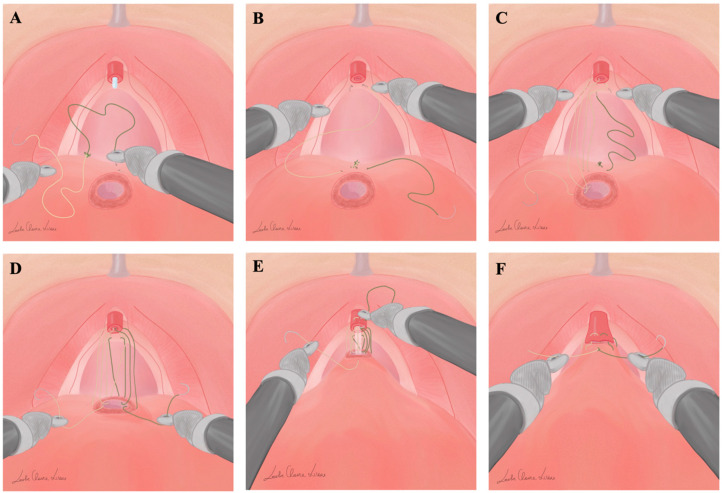
Key steps of Gallucci’s technique for vesicourethral anastomosis during RARP: the suture is passed outside-in on the bladder neck at the 6 o’clock position without including the bladder neck mucosa (**A**). The left suture is passed transversally through the posterior musculofascial plate from right to left without including urethral mucosa (**B**). After completing three passages with the left suture, the right suture is passed transversally, from left to right, through the posterior musculofascial plate, without including urethral mucosa (**C**). When three passages on each side have been completed, gentle traction is exerted on each suture, simultaneously and alternately, configuring the posterior aspects of the anastomosis (**D**). Similarly to the left suture, the right suture in passed inside-out at the 2 o’clock position, and again outside-in on the urethra at the 11 o’clock position (**E**). After passing the right suture inside-out on the bladder neck at the 1 o’clock position, the sutures are progressively tightened, and finally the ends of the two sutures are tied to one another on the outside of the bladder, completing the anastomosis (**F**).

**Figure 2 jpm-13-01072-f002:**
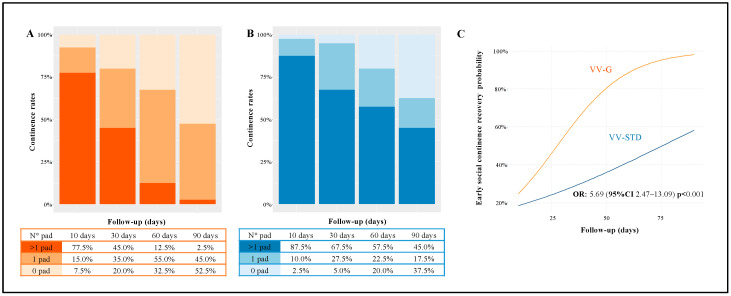
After 1:1 propensity score matching, continence rates stratified by 0 vs. 1 vs. >1 pads/die were reported at 10, 30, 60, and 90 days for Van Velthoven anastomosis modified sec. Gallucci (VV-G, (**A**)) vs. conventional Van Velthoven anastomosis (VV-STD, (**B**)). Based on generalized estimated equations (GEEs) for repeated measurements, the predicted probability of early social continence recovery during the first 90 days according to VV-G vs. VV-STD was depicted (**C**).

**Figure 3 jpm-13-01072-f003:**
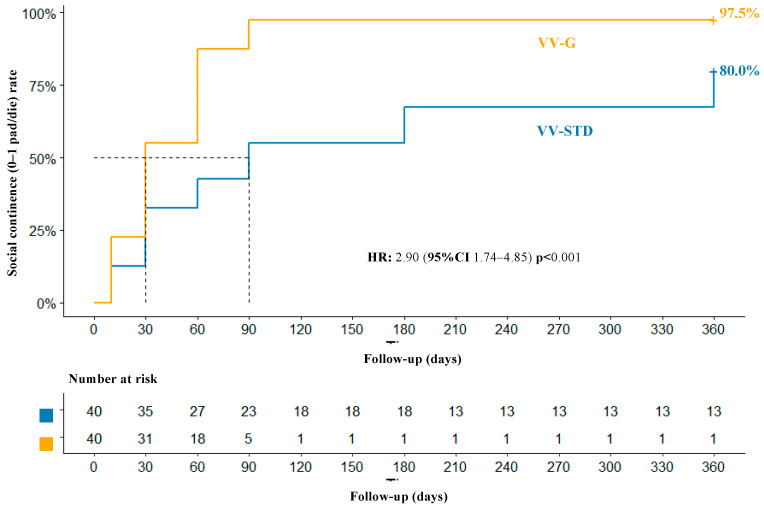
After 1:1 propensity score matching, a Kaplan–Maier plot and Cox regression model analyzed one-year overall social continence rate according to Van Velthoven anastomosis modified sec. Gallucci (VV-G) vs. conventional Van Velthoven anastomosis (VV-STD).

**Table 1 jpm-13-01072-t001:** Demographic and preoperative characteristics of the overall cohort according to VanVelthoven anastomosis modified sec. Gallucci (VV-G) vs. conventional Van Velthoven anastomosis (VV-STD) during RARP.

	VV-STD ^1^ *n* = 74 (45%)	VV-GALLUCCI ^1^ *n* = 92 (55%)	*p*-Value ^2^
** *Baseline characteristics* **			
**Age** (yr)	65.0 (62.0, 68.0)	66.0 (59.8, 70.0)	0.6
**BMI** (kg/m^2^)	27.1 (25.3, 29.4)	25.9 (24.1, 27.7)	0.026
**Smoking History**			0.7
*Never*	38 (51.4%)	44 (47.8%)	
*Former*	26 (35.1%)	31 (33.7%)	
*Current*	10 (13.5%)	17 (18.5%)	
**Diabetes**	10 (13.5%)	8 (8.7%)	0.3
**Charlson Comorbidity Index**	3 (2, 4)	3 (2, 4)	0.017
**Prostate volume** (cc)	39 (26, 55)	36 (25, 46)	0.3
**PSA** (ng/mL)	7.1 (5.0, 9.2)	7.6 (5.7, 11.0)	0.2
**ISUP grade biopsy**			0.7
*1*	16 (21.6%)	19 (20.7%)	
*2*	35 (47.3%)	42 (45.7%)	
*3*	13 (17.6%)	18 (19.6%)	
*4*	10 (13.5%)	10 (10.9%)	
*5*	0 (0.0%)	3 (3.3%)	
** *Perioperative characteristics* **			
**Nerve-Sparing**			<0.001
*No*	55 (74.3)	39 (42.4)	
*Monolateral*	6 (8.1)	16 (17.4)	
*Bilateral*	13 (17.6)	37 (40.2)	
**ePLND**	16 (21.6)	36 (39.1)	0.006
** *Pathologic report* **			
**Pathological Tumor Stage**			0.6
*pT2*	42 (56.8)	56 (60.8)	
*pT3a*	25 (33.8)	25 (27.2)	
*pT3b*	7 (9.5)	11 (12.0)	
**Pathological Nodes Stage**			0.043
*pNx*	58 (78.4)	56 (60.9)	
*pN0*	14 (18.9)	29 (31.5)	
*pN+*	2 (2.7)	7 (7.6)	
**Pathologic ISUP**			0.6
*1*	13 (17.6)	19 (20.7)	
*2*	38 (51.4)	40 (43.5)	
*3*	15 (20.3)	27 (29.3)	
*4*	3 (4.1)	2 (2.2)	
*5*	5 (6.8)	4 (4.3)	
**Positive SRM**	14 (18.9)	40 (43.5)	<0.001

BMI = body mass index; PSA = prostate-specific antigen; ePLND = extended pelvic lymph node dissection; SRM = surgical. ^1^ Median (IQR); *n* (%). ^2^ Wilcoxon rank sum test; Pearson’s Chi-square test.

## Data Availability

Department of Maternal-Child and Urological Sciences, Sapienza University Rome, Policlinico Umberto I Hospital, Rome, Italy; email: eugenio.bologna@uniroma1.it.

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
