# Peer review of "“Single Knot–Single Running Suture” Vesicourethral Anastomosis with Posterior Musculofascial Reconstruction during Robot-Assisted Radical Prostatectomy: A Step-by-Step Guide of Surgical Technique"

_jpm, 2023, doi:10.3390/jpm13071072_

Round 1

Reviewer 1 Report

Dear authors,

Congratulations on you effort and well written manuscript.

However, I have some questions.

First, the manuscript lacks ethical approval section (Informed consent, institutional review board approval etc). Since this is a prospective study, there should be one, otherwise it is unethical. One of the points in your discussion regarding the soundness of the study is the prospective design. If it was retrospective, than you wouldn't need informed consent.

Second, the study design is the major issue. Prospective, yet propensity score matched. There is a huge discrepancy in the rate of nerve sparing between VV-G and VV-STD, more nerve sparing was performed with VV-G which immediately affects the continence results. This is a major issue that is not discussed enough and that is why the conclusion can be that strong in favor of VV-G.

Third, I agree that it is an easy-to-learn and easy-to-teach technique, but what is the difference between some other posterior reconstruction techniques such as Rocco stich?

Fourth, you mention binding together two sutures for this technique. Have you tried double-needled single suture?

Reviewer 2 Report

The authors present a modification of the VV-technique including the posterior reconstruction in the same suture.

The technique is easy to perform. However, there are some concerns:

1. There is no citation of Rocco F or Rocco B, who introduced the concept of posterior reconstruction.

2. The technique of posterior reconstruction should be better described:

It is the anastomosis of the prostate-vesical muscle to the recto-urethralis muscle.

3. It might be clear, that the posterior reconstruction may lead to better early continence, .but this can also be done by separate sutures according to Rocco B et al. or Patel V et al. This needs to be discussed.

None

Round 2

Reviewer 1 Report

Dear authors,

Thank you for the revision of your manuscript.

Best regards.